# A Geometric Deep Learning Framework for Generation of Virtual Left Ventricles as Graphs

**Soodeh Kalaie**[*1]                                          S.KALAIE1@LEEDS.AC.UK

**Andy Bulpitt**[2]                                          A.J.BULPITT@LEEDS.AC.UK

**Alejandro F. Frangi**[3,4,5,6]                              A.FRANGI@LEEDS.AC.UK

**Ali Gooya**[7]                                          ALI.GOOYA@GLASGOW.AC.UK

[1] *Centre for Computational Imaging and Simulation Technologies in Biomedicine (CISTIB), School of Computing, University of Leeds, Leeds, UK*

[2] *School of Computing, University of Leeds, Leeds, UK*

[3] *Centre for Computational Imaging and Simulation Technologies in Biomedicine (CISTIB), School of Computing and School of Medicine, University of Leeds, Leeds, UK*

[4] *NIHR Leeds Biomedical Research Centre (BRC), Leeds, UK*

[5] *Alan Turing Institute, London, UK*

[6] *Medical Imaging Research Center (MIRC), Electrical Engineering and Cardiovascular Sciences Departments, KU Leuven, Leuven, Belgium*

[7] *School of Computing Science, University of Glasgow, Glasgow, UK*

**Editors:** Accepted for publication at MIDL 2023

## Abstract

Generative statistical models have a wide range of applications in the modelling of anatomies. In-silico clinical trials of medical devices, for instance, require the development of virtual populations of anatomy that capture enough variability while remaining plausible. Model construction and use are heavily influenced by the correspondence problem and establishing shape matching over a large number of training data. This study focuses on generating virtual cohorts of left ventricle geometries resembling different-sized shape populations, suitable for in-silico experiments. We present an unsupervised data-driven probabilistic generative model for shapes. This framework incorporates an attention-based shape matching procedure using graph neural networks, coupled with a $\beta-$VAE generation model, eliminating the need for initial shape correspondence. Left ventricle shapes derived from cardiac magnetic resonance images available in the UK Biobank are utilized for training and validating the framework. We investigate our method's generative capabilities in terms of generalisation and specificity and show that it is able to synthesise virtual populations of realistic shapes with volumetric measurements in line with actual clinical indices. Moreover, results show our method outperforms joint registration-PCA-based models.

**Keywords:** Attention, Generative modelling, Geometric Deep Learning, Graph Neural Networks, Shape matching, Virtual populations.

---

[*]Corresponding author

## 1. Introduction

Modelling cardiac anatomy using generative statistical models can have many applications, including identifying diseases, predicting them, and generating population cohorts for electrophysiological and mechanical computer simulations. Machine learning methods combined with computational modelling and simulation bring the possibility of gaining valuable information about new therapies and medical devices through In-Silico Clinical Trials (ISCTs) (Viceconti et al., 2016; Pappalardo et al., 2019). Therefore, virtual populations of anatomical shapes (typically represented as computational meshes) are a key enabler for conducting ISCTs of clinical devices.

However, building rich or descriptive generative shape models from inconsistent anatomical structures is challenging for several reasons. First, due to the fact that real-world anatomical shapes derived from different subjects do not generally share any topological correspondence, this is a challenging task for existing techniques to generate coherent, anatomically plausible shape populations. In addition, most techniques demand access to large volumes of training data, thus the process would require expensive and laborious annotation of medical imaging data. Also, it can still be computationally challenging to create synthetic samples that are clinically meaningful and that fully represent the characteristics of each individual patient. Therefore, this study aims to address the problem of generating virtual patient cohorts of left ventricle (LV) geometries resembling different-sized shape populations. These models allow to perform in-silico clinical trials on the so-called digital twins (Corral-Acero et al., 2020).

While PCA-based statistical shape models have been extensively used for shape generation (Piazzese et al., 2017; Gooya et al., 2017; Cosentino et al., 2020), deep learning approaches for generative modelling have gained increasing attention in recent years (Harshvardhan et al., 2020). A few studies have adopted these approaches for generating virtual populations of anatomy. Presenting cardiac biventricular anatomy as a point cloud, a geometric deep learning method is proposed for generating populations of realistic biventricular anatomies in (Beetz et al., 2021). The generation of personalized anatomies is further enhanced by adding subpopulation-specific characteristics as conditional inputs. Using binary masks of aortas as inputs, Romero *et al.* (Romero et al., 2021) explored the generation efficiency of the generative adversarial network (GAN) (Goodfellow et al., 2014). This method specifically addressed the generation of a cohort of patients meeting a specific clinical criterion. Danu *et al.* (Danu et al., 2019) employed deep generative models (VAE (Kingma and Welling, 2013) and GAN) for generating voxelised vessel surfaces, where they represented the unstructured surface mesh as a three-dimensional (3D) image. Although the results show potential for employing deep generative models on 3D surfaces, these models can not deal with complex data structures like bifurcations. In conclusion, firstly, these methods are limited by requiring a tedious pre-processing step to obtain dense point correspondence between training shapes. Secondly, they can only be applied to data with the same shape structure. Hence, presenting a meaningful shape matching (i.e. shape correspondence) across training data is often a prerequisite for these methods.

A number of problems can be categorized under the scope of shape correspondence. However, a unified approach can be achieved by considering the problem statement: given input shapes $\{g_k\}_{k=1}^K$, establish a meaningful relation $\mathcal{F}$ between their elements (such as

points, feature points, etc). The correspondence can be obtained directly based on the similarity of the elements, or it can be derived from the proximity of the aligned elements by aligning the shapes first. Additionally, we can iterate between the two procedures.

In the first scenario, the relation $\mathcal{F}$ is derived from aligning the shapes in a rigid or non-rigid manner. Some statistical approaches align two point sets via their Gaussian mixture representations (Myronenko and Song, 2010; Ma et al., 2020). Other approaches are taken in (Rusinkiewicz and Levoy, 2001; Zhang et al., 2021) for aligning point sets using optimising local quadratic distances. Moreover, due to the complexity of deformations in the optimisation process, the optimisation can easily get stuck in local optima. In another group, similarity-based methods estimate a pairwise assignment between shapes or their collected features to derive correspondences. More recent works on graph matching are based on deep learning methods to find the optimal point-to-point correspondences (Wang et al., 2019; Zanfir and Sminchisescu, 2018). These approaches develop supervised graph matching networks based on displacement rather than a registration task. However, the proposed graph matching methods are performed offline and remain unaltered during shape generation. Further, a pair-wised graph matching framework proposed in (Bai et al., 2018, 2019) is a learning-based framework rather than a pairwise graph distance computation. These methods employed multi-scale graph convolutional network (GCN) layers and then calculated multiple similarity matrices, increasing time complexity for large-scale graphs.

Compared with existing approaches, this study proposes an unsupervised geometric deep learning framework to generate virtual cohorts of left ventricle structures from different-sized training shapes. We consider the structures of the shapes within the context of the graphs. To the best of our knowledge, this is the first end-to-end deep learning framework capable of finding shape matching and generating anatomical shapes from different-sized shape populations without correspondence. The key contributions of this study are: (i) The framework presents a novel unsupervised learning-based shape matching procedure in the absence of any optimisation, which derives a learnable set of correspondences using graph neural networks (GNNs) and attention mechanisms. (ii) Using the soft attention mechanism, we present a domain transformation across the training data to address the problem of shape generation from different-sized shapes. (iii) We establish high-quality shape matching across the training data, and our non-linear generative framework is able to synthesise plausible LV shapes. Although the proposed generative framework is demonstrated here on left ventricle structures, it is generic in design and can be applied similarly to other anatomical shape ensembles.

## 2. Method

Our proposed framework consists of an unsupervised geometric deep learning network embedded in (i) an Attention-based Shape Matching (ASM) mechanism for deriving a learnable set of correspondences (ii) followed by a soft attention mechanism for domain transformation and $\beta$-VAE (Higgins et al., 2016) generation framework for synthesising 3D surface models of the shapes. Given a set of observed 3D surface meshes, we seek to develop a shape generative model using graph representation. Figure S1 shows the diagram of the proposed method.

## 2.1. Attention-based Shape Matching

Presenting the 3D surface mesh as a graph, we propose a matching procedure to determine correspondence based on the similarity measure of local node embeddings, without the need to solve an optimisation problem during inference. To compute nodal embeddings, GNNs enable us to efficiently and flexibly aggregate information through graph nodes and edges, which generate a powerful representation of shapes. Once the graph embedding is learned, an attention mechanism is applied to those embedded features to learn correspondence across the population.

Given a training set of different-sized graphs $G = \{g_k\}_{k=1}^K$, where $g_k = (V_k, E_k)$ is the $k$-th graph, with $|V_k|$ nodes, vertex Euclidean geometry feature matrix $\mathbf{X}_k \in \mathbb{R}^{|V_k| \times 3}$ and $E_k$ denotes the set of edges, connecting the vertices. The sparse adjacency matrix $\mathbf{A}_k \in \{0,1\}^{|V_k| \times |V_k|}$ represents the edge connections. In order to determine where and what shape matching should focus on, an attention mechanism is utilized. We model our attention-based matching procedure by computing similarities between nodal embeddings in the training and template graphs.

First, a variational graph auto-encoder network $\Psi$ (parameterized by $\{\boldsymbol{\theta}, \boldsymbol{\theta'}\}$) is employed to capture both local and global structural information in the shapes, where the encoder network ($\Psi_{\boldsymbol{\theta}}$) computes the nodal embedding vectors in the latent space. These embedding vectors are then used to compute and the pair-wise correspondences across the vertices of the $k$-th observed graph $g_k$ and the template graph $g_t$. Given the latent embeddings of nodes $\mathbf{Z}_k = \Psi_{\boldsymbol{\theta}}(\mathbf{X}_k, \mathbf{A}_k)$ and $\mathbf{Z}_t = \Psi_{\boldsymbol{\theta}}(\mathbf{X}_t, \mathbf{A}_t)$, computed by a shared network $\Psi$ for the observed graph and the template graph, respectively, we obtain the soft correspondence (i.e. attention maps), as the mapping function in the embedded-space paradigm

$$\mathbf{C}_k = Softmax(\lambda \mathbf{Z}_t \mathbf{Z}_k^T). \tag{1}$$

Where $\mathbf{Z}_k \in \mathbb{R}^{|V_k| \times d_z}$, $\mathbf{Z}_t \in \mathbb{R}^{|V_t| \times d_z}$, $\mathbf{C}_k \in [0,1]^{|V_t| \times |V_k|}$ and hyper-parameter $\lambda$ sets empirically. $d_z$ denotes the dimension of the latent vector $\mathbf{z}$. In this study, we exploit the ability of spatial-based geometric deep learning methods to handle inconsistent graph populations, where the convolution is performed in local Euclidean neighbourhoods. While the spectral-based approaches are limited to fixed graph structures due to filtering in the spectral domain. We consider a variational graph auto-encoder network $\Psi$ as a spatial-based GNN to perform graph convolution locally on each node. That is, the convolution operator learns features from the preceding network layer to dynamically determine the association between filter weights and graph neighbourhood, rather than relying on static predefined local pseudo-coordinate systems. Subsequently, to normalise shapes structurally, an attention mechanism is employed for domain transformation.

## 2.2. Shape Generation

The soft correspondence matrix $\mathbf{C}_k$ is a map from the node function space $\mathcal{F}(\mathbb{R}^{|V_k|})$ to node function space $\mathcal{F}(\mathbb{R}^{|V_t|})$ thus allows for domain transformations. To focus more on the relevant features of the shape, soft attention is implemented. In soft attention, irrelevant areas are discredited by multiplying the corresponding shape features with a low weight. Hence, to obtain shapes in the template domain, the soft attention mechanism directly

passes node features vector $\mathbf{x} \in \mathcal{F}(g)$ along with the soft correspondences

$$\mathbf{x}'_j = \sum_{n=1}^{|V_k|} C_{jn}\mathbf{x}_n \; ; j = \{1, ..., |V_t|\} \tag{2}$$

where $\mathbf{x}' \in \mathcal{F}(g')$ represents the node features vector in the other domain. $\mathbf{x}'_j$ is $j$-th row of matrix $\mathbf{X}'$ and vector $\mathbf{x}_n$ presents $n$-th row of feature matrix $\mathbf{X}$. Each structurally normalised shape $g'_k$ is presented by matrix feature representation $\mathbf{X}'_k \in \mathbb{R}^{|V_t| \times 3}$. Finally, a generation network, designed in $\beta$-VAE framework, learns a probability density function from a set of $\{g'_k\}_{k=1}^K$ graphs, which allows us to generate cohorts of artificial shapes.

### 2.3. Loss Function

A hierarchical unsupervised framework Attention-based Shape Matching (ASM) and Generation (G) is trained by minimizing the cost function $\mathcal{L} = \mathcal{L}_{ASM} + \mathcal{L}_G$, where $\mathcal{L}_{ASM}$ and $\mathcal{L}_G$ in the right hand side are defined subsequently.

Shape matching cost: In the shape matching procedure, a refinement strategy is also considered in order to avoid finding false correspondences. To achieve this, the loss $\mathcal{L}_{ASM}$ is computed as

$$\mathcal{L}_{ASM} = \mathcal{L}_\Psi + \mathcal{L}_{Ref} = \underbrace{\sum_{k=1}^K \frac{1}{2}\sum_{n=1}^{|V_k|} \|\mathbf{x}_{kn}^{rec} - \mathbf{x}_{kn}\|^2 - w_0 D_{KL}}_{\mathcal{L}_\Psi} + \underbrace{w_1 CD(g'_k, g_k) + w_2 Lap(g'_k)}_{\mathcal{L}_{Ref}},$$

$$\tag{3}$$

where $\mathcal{L}_\Psi$ is minimised to learn nodal embeddings mentioned in section 2.1 (the details can be found in supp.mat 5.2). The loss term $\mathcal{L}_{Ref}$ (refers to the refinement strategy), consisting of Chamfer and Laplacian losses, is minimised to refine the structurally normalised shapes.

Chamfer loss measures the distance of vertices between two graphs: $CD(g, g') = \sum_{\mathbf{x} \in g} \min_{g'} \|\mathbf{x} - \mathbf{x}'\|^2 + \sum_{\mathbf{x}' \in g'} \min_g \|\mathbf{x} - \mathbf{x}'\|^2$ and $Lap(g'_k)$ is a laplacian smoothness loss to result in smoother surface reconstructions (Taubin, 1995). The weights $w_0, w_1$ and $w_2$ are hyperparameters that are tuned and set empirically.

Generation cost: The loss function $\mathcal{L}_G$ follows the original loss of the $\beta$-VAE, where hyperparameter $\beta$ makes a balance between low reconstruction error and high latent space quality, which emphasizes discovering disentangled latent factors. Assuming VAE is made of an encoder-decoder pair, we define our architecture as a pair $\{E, G\}$ neural networks, respectively. Following this, we define:

$$\mathcal{L}_G = \frac{1}{2}\sum_{k=1}^K \sum_{j=1}^{|V_t|} \left\| G(E(\mathbf{x}'_{kj})) - \mathbf{x}'_{kj} \right\|^2 - \beta D_{KL}, \tag{4}$$

where $D_{KL}$ denotes the Kullback-Leibler, which computes the divergence between the Gaussian prior $\mathcal{N}(0, I)$ and posterior distributions of the latent space $E(\mathbf{x}')$.

## 3. Experiments and Results

In this section, we describe the experimental settings and the evaluation of our method on the clinical dataset. **Dataset**: We utilized 1000 LV 3D surface mesh dataset obtained from cardiac magnetic resonance (CMR) images of the UK Biobank (UKB) (Petersen et al., 2015), using the pipeline described in (Zakeri et al., 2022). All the LV meshes are related to the end-diastolic phase of the cardiac cycle. Mesh cardinalities vary from 1000 to 2000 points. We randomly split the data into 720 training, 80 validation, and 200 test cases. ( In the supp. mat, additional experiments on a different dataset are provided to evaluate the model's versatility.) **Experimental Setup**: All experiments are carried out using PyTorch. We used the spatial graph convolution operator proposed in (Verma et al., 2018) to build the layers of the network $\Psi$ and the fully-connected layers are used in the structure of the $\beta$-VAE network for the shape generation. More details are provided in the supp. mat 5.3.

### 3.1. Results and Discussion

A number of experiments were conducted to evaluate the performance of the generative shape framework proposed in this study. Since existing works on shape generative modelling can only be applied to data with the same shape structure, we compared our model with a baseline model, where the pipeline includes registration and PCA models. We evaluate our method in terms of both its matching (i.e. correspondence establishment) and generation performances.

#### 3.1.1. Matching quality

We investigated the proposed Attention-based Shape Matching (ASM) framework in two settings, With/WithOut Refinement, and compared them with the rigid Registration-based Shape Matching (RSM) proposed in (Myronenko and Song, 2010). The correspondence maps obtained from different methods RSM and ASM(WoR/WR) are utilized to transform actual shapes $g_k$ from the domain $\mathbb{R}^{|V_k|}$ to $\mathbb{R}^{|V_t|}$. For a qualitative evaluation, Figure 1 visualises some examples of the network input ($g_k$) and normalised surface mesh ($g'_k$) for five sample cases, obtained from the different methods. Visual inspection of results shows that normalised shapes obtained from our framework (especially with the refinement strategy introduced in section 2.3) are more realistic and present meaningful correspondences for anatomical landmarks on LV shapes (e.g. endocardial, epicardial, LV base, and apex), whereas the results obtained from RSM show some disorders and lacking details. We further evaluated the quality of obtained shapes in the template domain using two distance metrics: Hausdorff distance ($HD$) and Chamfer distance ($CD$). Table 1 summarises the accuracy of shapes in the template domain obtained by different methods. Obtained results show that our similarity-based method maintains high accuracy when compared with the registration-based method. These results suggest that the refinement strategy produces more accurate features derived from the network $\Psi$ with a spatial-based graph convolutional layer, thus providing high-quality correspondences. By achieving lower mean Hausdorff and Chamfer distances, our method demonstrates good normalisation quality in arbitrary target domains, while the low standard deviation values demonstrate its robustness. This is because our method followed an efficient spatial-based geometric deep learning strategy and considered a learning-based fully-differentiable shape matching procedure that aimed

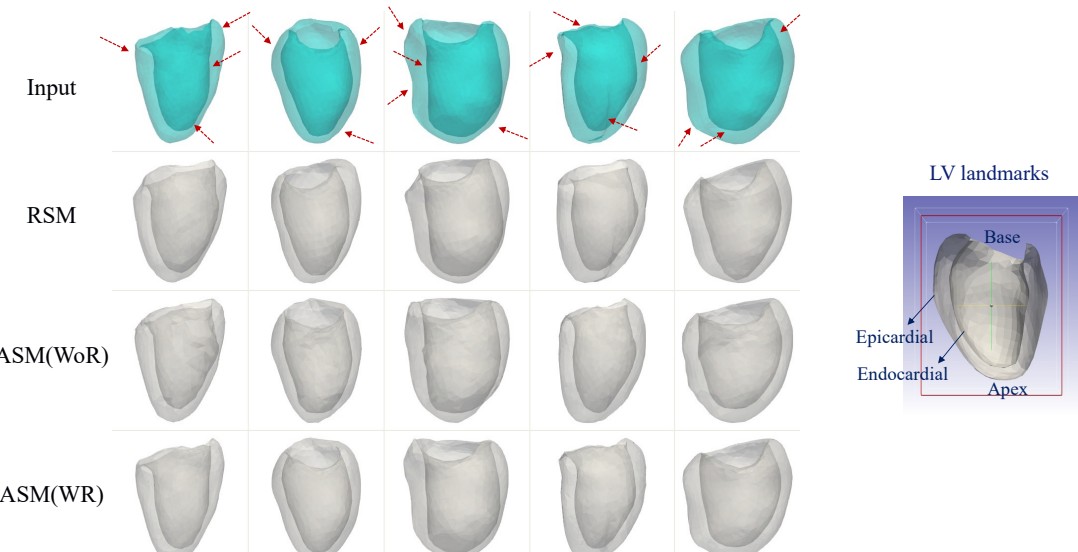

Figure 1: Examples show the comparison between different shape matching approaches for deriving correspondence. Cyan-coloured meshes present input $g_k$ with cardinality $|V_k| = 1039, 1216, 1353, 1497$ and $1503$ respectively from left to right. Grey-coloured shapes are the normalised meshes $g'_k$ with cardinality $|V_t| = 1093$ obtained from different methods. Notice endocardial, epicardial, LV base, and apex landmarks.

Table 1: Shape Matching Quality: comparison between different methods using two distance metrics $HD$ and $CD$ (mean $\pm$ std) in $[mm]$. **Bold** values show a significant difference between the methods with a p-value $< 0.001$ using the statistical paired t-test.

|  | RSM | ASM(WoR) | ASM(WR) |
|---|---|---|---|
| $HD$ | $8.11 \pm 2.13$ | $8.32 \pm 1.77$ | $\mathbf{6.54 \pm 1.57}$ |
| $CD$ | $12.04 \pm 2.63$ | $9.91 \pm 1.54$ | $\mathbf{4.06 \pm 0.57}$ |

to reach a data-driven neighbourhood between matched node pairs without the need to solve any optimisation.

### 3.1.2. Generation quality

In order to evaluate the generative performance of our framework (ASMG), we provide a quantitative assessment in terms of generalisation, specificity (Styner et al., 2003) and clinical relevance. The generalisation ability of a generative model indicates the capability of the model to represent unseen samples and thus capture the variability in LV shapes based on its error when reconstructing unseen actual test data. Model specificity determines the anatomical plausibility of the virtual LV cohorts by comparing them to the actual samples in the training set. In order to report generalisation and specificity, three distance measures are considered: Hausdorff distance ($HD$), the average of minimum Euclidean ($ED$) distance and

Table 2: Generation Quality: comparison between generative models in terms of generalisation ability and specificity using three distance metrics $HD$, $ED$ and $ED^*$ (mean $\pm$ std) in $[mm]$. Statistically significant (p-value $< 0.001$) performance of ASMG(WR) method over RSMP is shown by **Bold** values.

|  |  | RSMP | ASMG(WoR) | ASMG(WR) |
|---|---|---|---|---|
|  | $HD$ | $8.02 \pm 2.14$ | $7.99 \pm 1.83$ | $\mathbf{7.08 \pm 1.58}$ |
| Generalisation | $ED$ | $2.15 \pm 0.20$ | $2.11 \pm 0.14$ | $\mathbf{2.05 \pm 0.13}$ |
|  | $ED^*$ | $2.38 \pm 0.24$ | $2.36 \pm 0.17$ | $\mathbf{2.23 \pm 0.11}$ |
|  | $HD$ | $5.96 \pm 0.59$ | $5.82 \pm 0.70$ | $\mathbf{5.80 \pm 0.99}$ |
| Specificity | $ED$ | $2.28 \pm 0.22$ | $2.18 \pm 0.13$ | $\mathbf{2.25 \pm 0.20}$ |
|  | $ED^*$ | $2.23 \pm 0.28$ | $2.47 \pm 0.18$ | $\mathbf{2.55 \pm 0.20}$ |

Table 3: Clinical acceptance rates $\mathcal{A}$ [in %] achieved by each generative model for LV volumes.

|  | RSMP | ASMG(WoR) | ASMG(WR) |
|---|---|---|---|
| $\mathcal{A}_{[min,max]}$ | 93.95 | 100 | 100 |
| $\mathcal{A}_{M \pm 2B}$ | 47.73 | 88.88 | 91.62 |
| $\mathcal{A}_{\mu \pm \sigma}$ | 24.38 | 56.50 | 62.82 |

its symmetric distance ($ED^*$), where $ED^*(g, g') = ED(g', g)$. High-performance generative models refer to models that generate synthetic shapes with simultaneously low specificity and generalisation errors. Table 2 summarises the generalisation and specificity errors of all methods investigated in this study (ASMG(WoR) and ASMG(WR)) and the baseline model RSMP, where the generative model RSMP is built from the synergy of Registration-based Shape Matching and PCA generator. We observe that the ASMG model achieves the highest generalisability and specificity (i.e. lower concurrent specificity and generalisation errors for the majority of distances) for both investigated scenarios. As expected, due to lower specificity errors, our similarity-based generative model presents more realistic synthesised LV shapes when compared with the model based on registration. The lower specificity error can be explained as follows: considering shapes as graphs support our unsupervised shape matching framework to learn better disentangled latent representations and thus derive a more effective form of soft correspondences between shapes, which in turn preserves more details during the normalisation process. As a result, shapes generated from structurally normalised populations have a greater degree of plausibility. Compared to PCA, which is a linear projection of shapes onto lower-dimensional subspaces, our generative model is based on graph convolution networks and $\beta-$VAE, which can capture non-linear variations in shapes. A proper balance between latent space and reconstruction quality is achieved by using the $\beta-$VAE generator, resulting in realistic randomly generated samples.

Inspired by (Romero et al., 2021), we present the clinical relevance assessment for the synthesised LV shapes in this study. Given the actual cohort UKB, the acceptance rate $\mathcal{A}$

determines the percentage of synthetic samples in virtual cohorts with cardiac indices (e.g. volume), within a confidence interval of the distribution of the cardiac indices observed in the actual population. To compute the acceptance rate $\mathcal{A}$, three different confidence intervals are considered. First, $[\min, \max]$ interval presents the range of the observed actual biomarkers (e.g. LV volume). Relying on Chebyshev's inequality, we define two confidence intervals $\mu \pm \sigma$ and $M \pm 2B$, based on the corresponding mean ($\mu$), standard deviation ($\sigma$) and the mode ($M$) observed in the actual population (see Figure S3). Where $B = \sqrt{\sigma^2 + (M - \mu)^2}$ measures the variability across the data. Table 3 shows the efficiency of each method measured using the different acceptance criteria. The clinical acceptance rates of LV virtual cohorts synthesised by our generative model are higher than RSMP method. Although the specificity errors in Table 2 indicate that there is no huge difference between the plausibility of LV shapes synthesised by the model with and without refinement strategy, the clinical acceptance rates estimated for ASMG with refinement are consistently higher than those without refinement across the LV volume indices (refer to Table 3). This indicates that the generative model proposed in this study, when considering the refinement strategy, provides better fidelity in preserving the distributions of clinically relevant cardiac indices in the synthesised virtual cohorts relative to the UKB population. It is therefore more suitable for ISCTs that require higher statistical fidelity based on anatomical characteristics.

## 4. Conclusion

This study proposed a novel unsupervised probabilistic deep generative model, capable of generating virtual LV shape cohorts, resembling the different-sized actual data. The framework is a synergy of attention-based graph convolutional networks and generator modules which do not require any prior landmarking, making it significantly efficient. Furthermore, this method is capable of working on 3D image-derived surface meshes due to its use of graphs instead of highly sparse and memory-intensive voxel grids. Additionally, using spatial-based graph convolutional networks in the matching procedure made the proposed method well-suited for handling inconsistent shape structures. The study demonstrated the suitability and applicability of the proposed generative model in conducting ISCTs, with virtual cohorts, through a comparative analysis. Further extensions to our work would include richer descriptions of vertex geometry, such as normal vertex along the vertex geometry, and different strategies for sampling generation.

## Acknowledgments

The UKB CMR dataset has been provided under UK Biobank access application number 11350.

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

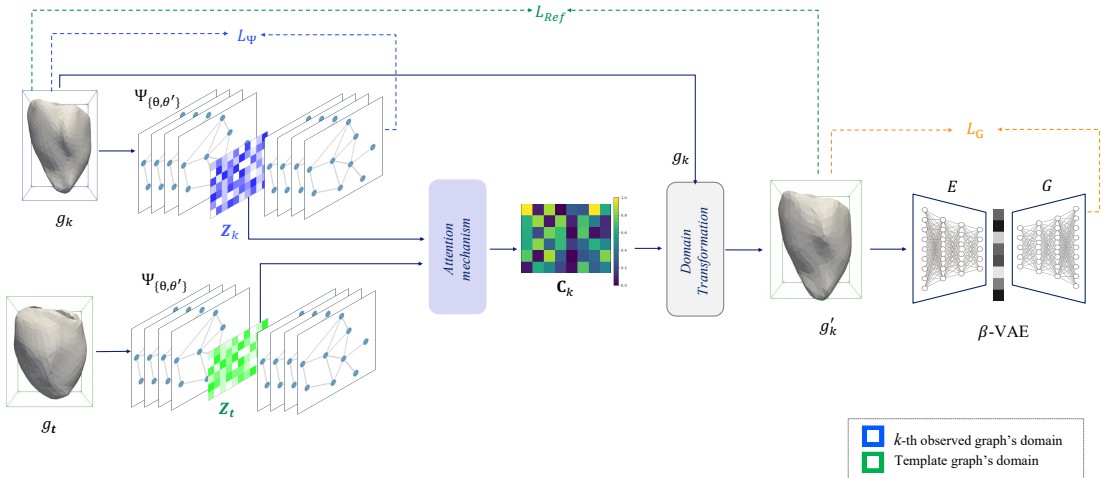

Figure S1: Overview of the ASMG framework for shape generation from different-sized shapes.

## 5. Supplementary Material

In this supplementary document, we first provide an overview of the proposed hierarchical unsupervised framework of attention-based shape matching and generation (ASMG). Subsequently, we provide the details on the variational graph autoencoder network and its loss function. Afterward, we introduce the implementation details of our method. Eventually, we also show additional experiments of our method.

### 5.1. Overview of Our Unsupervised Learning Approach (ASMG)

Figure S1 shows a diagram of the proposed ASMG method, where a hierarchical framework is trained to generate shape structures by minimizing the cost function mentioned in section 2.3. First, a synergy of variational graph convolutional network $\Psi$ and attention mechanism establishes vertex-to-vertex correspondences ($\mathbf{C}$) between the shapes in the latent space. Subsequently, a variational autoencoder ($\beta$-VAE) learns a probability density function from a set of structurally normalized shapes ($g'$) in the 3D space.

### 5.2. Variational Graph Autoencoder Network $\Psi$

Variational graph autoencoder network $\Psi$ takes the adjacency matrix $\mathbf{A}$ and node features $\mathbf{X}$ as input and tries to reconstruct the feature matrix $\mathbf{X}$ (refers as $\mathbf{X}^{rec}$) through the hidden layer embeddings $\mathbf{Z}$. Let us parameterise the approximate posterior $q_{\boldsymbol{\theta}}(\mathbf{Z}|\mathbf{X}, \mathbf{A})$ with an encoder and the likelihood $p_{\boldsymbol{\theta}'}(\mathbf{X}|\mathbf{Z}, \mathbf{A})$ with a decoder. The network is trained in an unsupervised manner by maximising the evidence lower bound (ELBO) $\mathcal{L}_{\Psi}$ w.r.t. the

variational parameters:

$$\mathcal{L}_\Psi(\boldsymbol{\theta}, \boldsymbol{\theta}') = \sum_{k=1}^{K} (\mathbb{E}_{q_{\boldsymbol{\theta}}(\mathbf{Z}_k|\mathbf{X}_k, \mathbf{A}_k)}[\log p_{\boldsymbol{\theta}'}(\mathbf{X}_k|\mathbf{Z}_k, \mathbf{A}_k)] - w_0 D_{KL}[q_{\boldsymbol{\theta}}(\mathbf{Z}_k|\mathbf{X}_k, \mathbf{A}_k) \parallel p(\mathbf{Z}_k)])$$

$$= \frac{1}{2} \sum_{k=1}^{K} \sum_{i=1}^{|V_k|} \|\mathbf{x}_{ki}^{rec} - \mathbf{x}_{ki}\|^2 - w_0 D_{KL}[q_{\boldsymbol{\theta}}(\mathbf{Z}_k|\mathbf{X}_k, \mathbf{A}_k) \parallel p(\mathbf{Z}_k)], \tag{5}$$

where the expected likelihood term interprets as a reconstruction loss and measures the squared Euclidean distance between real and reconstructed shapes by the decoder. $D_{KL}$ is the Kullback-Leibler divergence (KL divergence) between the approximate posterior $q_{\boldsymbol{\theta}}(\mathbf{Z}_k|\mathbf{X}_k, \mathbf{A}_k)$ and the prior distribution $p(\mathbf{Z}_k)$, weighted by $w_0$. Unit Gaussian distribution defines a prior distribution $p(\mathbf{Z}_k) = \prod_{n=1}^{|V_k|} N(\mathbf{z}_{kn}; \mathbf{0}, \mathbf{I})$.

The inference model parameterized by graph convolutional layers (GCN):

$$q_\theta(\mathbf{Z}_k|\mathbf{X}_k, \mathbf{A}_k) = \prod_{n=1}^{|V_k|} q(\mathbf{z}_{kn}|\mathbf{X}_k, \mathbf{A}_k)$$

$$\text{with} \quad q(\mathbf{z}_{kn}|\mathbf{X}_k, \mathbf{A}_k) = N(\mathbf{z}_{kn}|\boldsymbol{\mu}_{kn}, diag(\boldsymbol{\sigma}_{kn}^2)) \tag{6}$$

where $\boldsymbol{\mu} = \mathrm{GCN}_{\boldsymbol{\mu}}(\mathbf{X}, \mathbf{A})$ is the matrix of mean vectors $\boldsymbol{\mu}_n$ and $\log\boldsymbol{\sigma} = \mathrm{GCN}_{\boldsymbol{\sigma}}(\mathbf{X}, \mathbf{A})$.

### 5.3. Experimental Setup

The network $\Psi$ uses the spatial-based GCN layers proposed in (Verma et al., 2018), with $64, 64, 128, 128$-dim hidden layers for the encoder (which are mirrored for the decoder) with convolutional filter weight matrices of size 8. The structure of this network is also shown in Table S1. More specifically, inspired by (Verma et al., 2018) we use the spatial-based graph convolution operator (named feature-steered convolutional operator) where the operator dynamically assigns filter weights to the node's neighbourhoods according to the features learned by the network.

To build the shape generation network fully-connected layers are used in the structure of the $\beta$-VAE network with hidden layers of size $512, 256, 768, 128, 64$ for the encoder. The decoder is a mirrored version of the encoder. All the internal layers use batch normalisation and Leaky ReLU as activation layers. We empirically set hyperparameters $\gamma$, $w_0$, $w_1$, $w_2$ and $\beta$ to $10^4$, $1e^{-3}$, $1$, $1$ and $2e^{-6}$ respectively. We set the learning rate to $1e^{-3}$ and use the ADAM optimiser (Kingma and Ba, 2014) to train the model on LV dataset. A canonical shape has been randomly selected as the template graph $g_t$.

### 5.4. Additional Experiments

Figure S2 illustrates how the samples from Gaussian prior $\mathcal{N}(0, I)$ match learned posterior distributions in $\beta-$VAE, for eight test LV shapes. In all cases, we get a p-value$> 0.001$ which implies that we do not reject the null hypothesis that the distribution of the posterior is the same as prior.

The resulting LV volume indices distributions are presented in Figure S3 by means of violin charts. As shown in this figure, horizontal lines indicate the boundaries of the different

Table S1: The structure of network $\Psi$, where $|V|$ denotes the cardinality of input shape.

| Layer | Filter weight matrix size | Input size | Output size |
| --- | --- | --- | --- |
| Spatial GCN | 8 | $|V| \times 3$ | $|V| \times 64$ |
| Batch Norm | - | $|V| \times 64$ | $|V| \times 64$ |
| Spatial GCN | 8 | $|V| \times 64$ | $|V| \times 64$ |
| Batch Norm | - | $|V| \times 64$ | $|V| \times 64$ |
| Spatial GCN | 8 | $|V| \times 64$ | $|V| \times 128$ |
| Batch Norm | - | $|V| \times 128$ | $|V| \times 128$ |
| Spatial GCN ($\boldsymbol{\mu}$) | 8 | $|V| \times 128$ | $|V| \times 128$ |
| Spatial GCN ($\log\boldsymbol{\sigma}$) | 8 | $|V| \times 128$ | $|V| \times 128$ |
| Spatial GCN | 8 | $|V| \times 128$ | $|V| \times 128$ |
| Batch Norm | - | $|V| \times 128$ | $|V| \times 128$ |
| Spatial GCN | 8 | $|V| \times 128$ | $|V| \times 64$ |
| Batch Norm | - | $|V| \times 64$ | $|V| \times 64$ |
| Spatial GCN | 8 | $|V| \times 64$ | $|V| \times 64$ |
| Batch Norm | - | $|V| \times 64$ | $|V| \times 64$ |
| Spatial GCN | 8 | $|V| \times 64$ | $|V| \times 3$ |

Kernel Density Function

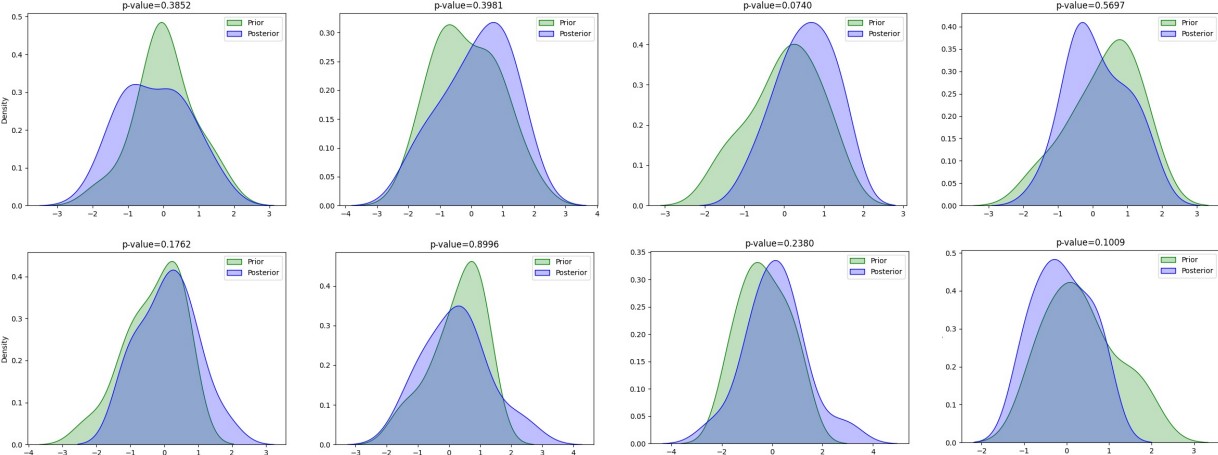

Figure S2: Visualization of how the samples from Gaussian prior $\mathcal{N}(0, I)$ match learned posterior distributions in $\beta-$VAE, for eight test LV shapes.

acceptance criteria. Comparing actual to synthetic distributions for volume variables show that our model generates more realistic samples from the population of LV while capturing sufficient variability. PCA-based generative model RSMP, with higher generalisation and specificity errors, is reflected in the wider range (unrealistic) of values observed for volume

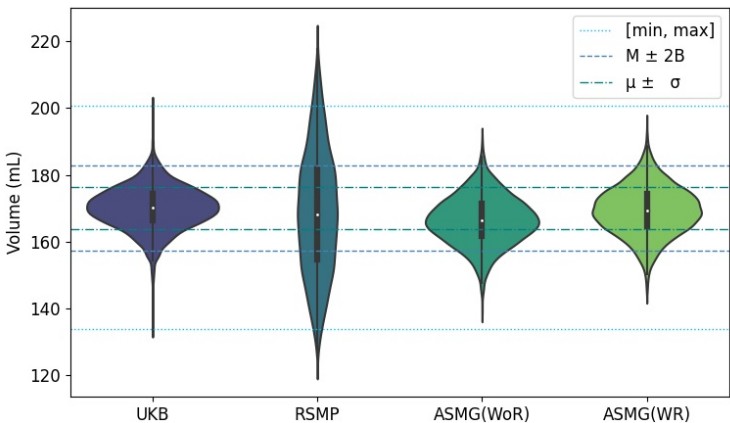

Figure S3: Violin plots for the distribution of LV volume indices on the actual UKB samples, alongside with those generated using methods: RSMP, ASMG(WoR) and ASMG(WR).

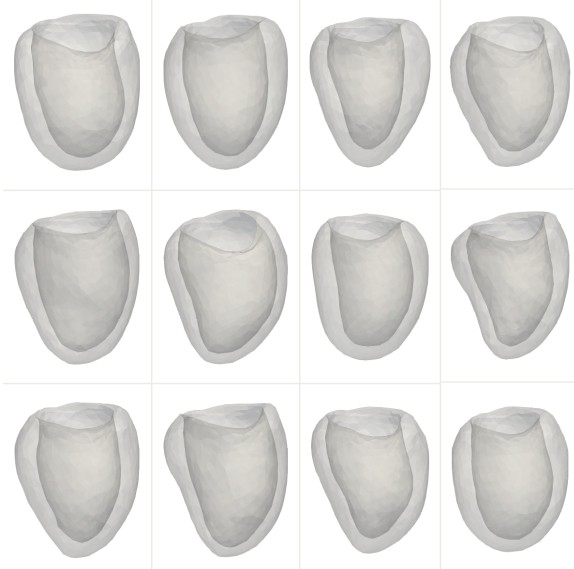

Figure S4: Examples of virtual samples generated by the ASMG(WR) generator model.

indices in the synthesised virtual population. In addition, a visualization of the generated (synthetic) samples by the trained ASMG(WR) model is shown in Figure S4.

### 5.4.1. Model's Versatility

To evaluate the model's versatility, another dataset (i.e. liver dataset) with a different number of shapes and different complexities in the structure is used. Compared to the LV dataset, the liver dataset demonstrates a wider volumetric variation. Additionally, in

Table S2: Generation Quality on the Liver dataset: comparison between generative models in terms of generalisation ability and specificity using three distance metrics $HD$, $ED$ and $ED^*$ (mean $\pm$ std) in $[mm]$. Statistically significant (p-value < 0.001) performance of ASMG(WR) method over RSMP is shown by **Bold** values.

|  |  | RSMP | ASMG(WoR) | ASMG(WR) |
|---|---|---|---|---|
| | $HD$ | $35.94 \pm 14.6$ | $32.44 \pm 6.86$ | $\mathbf{32.86 \pm 5.44}$ |
| Generalisation | $ED$ | $7.72 \pm 2.39$ | $8.09 \pm 1.56$ | $\mathbf{7.56 \pm 1.20}$ |
| | $ED^*$ | $10.51 \pm 3.83$ | $9.98 \pm 1.44$ | $\mathbf{9.50 \pm 1.35}$ |
| | $HD$ | $29.60 \pm 5.94$ | $24.83 \pm 1.80$ | $\mathbf{24.13 \pm 2.94}$ |
| Specificity | $ED$ | $8.41 \pm 1.50$ | $6.60 \pm 0.43$ | $\mathbf{6.93 \pm 0.60}$ |
| | $ED^*$ | $8.42 \pm 2.16$ | $6.50 \pm 1.68$ | $\mathbf{8.28 \pm 0.87}$ |

Table S3: Clinical acceptance rates $\mathcal{A}$ [in %] achieved by each generative model for liver volumes.

|  | RSMP | ASMG(WoR) | ASMG(WR) |
|---|---|---|---|
| $\mathcal{A}_{[\min,\max]}$ | 64.03 | 100 | 100 |
| $\mathcal{A}_{M\pm2B}$ | 46.76 | 100 | 100 |
| $\mathcal{A}_{\mu\pm\sigma}$ | 19.42 | 90.65 | 79.14 |

general, there is further morphological variability across the liver shapes, which imposes more challenges when training the generative model.

Liver dataset: 3D liver shapes are obtained from the public CT-ORG dataset from The Cancer Imaging Archive (TCIA). To prepare our required training graph dataset, 3D surface liver meshes are reconstructed from CT scans of 139 patients using the MarchingCube algorithm. To train the model on the liver dataset, we empirically set hyperparameters $\gamma$, $w_0$, $w_1$, $w_2$ and $\beta$ to $10^4$, $1e^{-3}$, 1, 1.2 and $2e^{-3}$ respectively.

The results in Table S2 show that the average distances in specificity and generalisation metrics are significantly lower for ASMG(WR) model compared to RSMP, (p-value< 0.001) and ASMG(WR) outperforms RSMP. The lower values in specificity errors indicate that the accuracy of newly generated instances by the generative RSMG(WR) model is higher and this model presents more realistic synthetic shapes in the liver dataset. In the majority of the average distances, the ASMG(WR) model outperforms the ASMG(WoR) and indicates smaller concurrent specificity and generalization errors.

Table S3 summarises the clinical acceptance rates for the liver volumes synthesised by different generative models. ASMG(WoR/WR) obtain higher acceptance rates across the liver volume indices, which is consistent with the specificity summarised in Table S2. Although the clinical acceptance rate $\mathcal{A}_{\mu\pm\sigma}$ estimated for ASMG(WoR) is higher than ASMG(WR) generator model, the specificity errors indicate that the ASMG(WR) model synthesises liver shapes that are more plausible. This model is therefore more suitable for ISCTs that require a higher degree of statistical fidelity based on liver anatomical characteristics.

In conclusion, the results of the study indicate that the performance of the proposed method is minimally affected by the morphological variability of liver anatomy.

