# OpenReview forum: "A Geometric Deep Learning Framework for Generation of Virtual Left Ventricles as Graphs"
_MIDL.io/2023/Conference — MIDL 2023 Oral_

### Official Review · Reviewer_E9Ho · 2023-02-02

**Confidence:** 3
**Preliminary Rating:** 3

**Summary:**

The main objective of this paper is to present a novel method for generatiing virtual left ventricles. For that matter, it combines an unsupervized method for shape matching using GNNs and attention, a domain transformation to normalize shape structure and finally $\beta$-VAE for generation. This original end-to-end approach for generating LV shapes is then evaluated for quality with comparison with a classical model and for its generalisation ability by using an interesting method from recent literature to assess clinical relevance.

**Strengths:**

- The problem is relevant and has no satisfactory solution in the literature yet. Positioning w.r.t. sota is clear.
- The main contributions of this work (attention based matching and end-to-end framework) are novel and the results presented are convincing.
- Methodology is sound.
- The paper is globally well-written.

**Weaknesses:**

- The paper is sometimes vague on points that would have deserved a longer discussion especially on 2.1 on attention shape matching that is the key contribution of the paper. As is, all steps are not completely clear to me (SpaGNN...).
- Reproducibility is a bit weak:
     - some important details (exact network structures...) are given as references to other papers (e.g. FeaStNet). For the sake of reader's convenience, they should at least be given in supplementary materials.
    - selection/tuning method for hyperparameters, seeds, training/validation splits and their influence on the outcome are not described. As such, the training procedure cannot be reproduced?




**Deanonymize Review:**

no

**Detailed Comments:**

Some typos should be corrected eg 'point-to-pint' in page 2, "Exprimental Setup" in page 5, "," after equations if they are followed by "Where".


**Paper Type:**

both

**Questions To Address In The Rebuttal:**

The main question to be addressed concerns the reproducibility of results. As the diffusion of a working code is not mentioned, one is expecting at least the complete description of the methodology and implementation details. Some may be given as supplementary material. Also related to reproducibility and generalization, the influence of uncertainties and empirical choices should be investigated to improve the evaluation of the generalization ability of the framework.

---

### Official Review · Reviewer_aLEc · 2023-02-04

**Confidence:** 2
**Preliminary Rating:** 4

**Summary:**

The authors present a generative model for generating left ventricles represented as graphs. The framework is  includes a GNN-based shape matching and a beta-VAE to generate samples, and is trained end-to-end. The method is validated on data from the UK biobank and shown to outperform a registration+PCA baseline model.

**Strengths:**

- The method is trained end-to-end to simultaneously perform shape matching and generation
- The method naturally incorporates different sized graphs without the need for standardisation
- The method outperforms the baseline

**Weaknesses:**

- The authors only validate on data from the UK Biobank. The paper would be strengthened by showing the method's performance generalises to other datasets.

- In a similar vein, the authors state the method will generalise to other structures, but don't demonstrate this.

**Deanonymize Review:**

no

**Detailed Comments:**

Beta-VAEs are outperformed by a whole host of generative models (in the imaging domain, at least). I'd be curious to learn if the authors considered any more recent generative models and why they settled on the beta-VAE?

In Table 2 it isn't clear if values aren't bolded because the authors choose not to test for statistical significant, or because none of the methods are better when tested. Please clarify.

Do the authors plan to make their code available?

The landmarks highlighted in Figure 1 aren't very easy to identify for those not versed in the clinical task. Explicit labelling of them on each image could help?

**Paper Type:**

methodological development

**Questions To Address In The Rebuttal:**

Why did the authors not validate on more ventricle datasets, from different sources, or different types of structure to demonstrate the generalisability of the method?

Why did the authors choose the Beta-VAE and did they experiment with other models?

---

### Official Review · Reviewer_Ddky · 2023-02-07

**Confidence:** 4
**Preliminary Rating:** 3
**Recommendation:** Poster

**Summary:**

The paper proposes a generative model for volumetric shapes, that utilizes a Beta Variational Auto-Encoder with additional regularizations to match an input shape to a template shape in the latent space of the $\beta$-VAE, therefore implicitly solving the shape correspondence problem in the latent space. In short, the input mesh for a shape is treated as a graph, and a graph convolutional autoencoder is used to process this graph. The node embeddings (output of the encoder) are calculated for an input graph and a template graph (i.e., mesh of a template shape), and the soft assignment between the nodes is calculated via normalized cross-correlation  (aka, attention!). The soft-assignment/normalized-cross-correlation matrix is used to map the input graph to the template graph, representing an approximation of the embedded with the same number of nodes as the template, $|V_t|$. Having a canonical representation (in the latent/embedding space) allows for generative modeling of the shapes by sampling from the prior distribution of the VAE. The authors numerically evaluate the proposed method on modeling left ventricle shapes.

**Strengths:**


* The problem that the paper addresses is of interest to the community and has practical applications.
* Geometric deep learning approaches for analyzing mesh data is a timely topic with diverse applications in medical image/data analysis.
* The proposed idea is interesting and has potential.
* While VAEs and AEs have been extensively applied for encoding shapes, the soft-alignment in the latent space through normalized cross-correlation (aka, attention) is interesting and a viable idea.



**Weaknesses:**

The paper has several shortcomings, which I enumerate below:

1. While the proposed ideas are interesting, they are presented in a convoluted and overly complicated manner. The presentation of the paper can be significantly improved (See my comments to the authors).

2. Section 2 and, in particular, `Section 2.3. Loss Function' is poorly written, with several mistakes in the formulation/notations.

3. Given the shortcomings above, I would have expected a more extensive experimental section to gain any confidence about the proposed approach.

**Deanonymize Review:**

no

**Detailed Comments:**

Below are the mistakes and inconsistencies in Section 2.3:
   * Typo: Before Eq (3): $\mathcal{L}\_{SM}$ should be $\mathcal{L}\_{ASM}$
   * It is not clear which part of Eq (3) corresponds to which of the two losses, $\mathcal{L}_{\Psi}$ and $\mathcal{L}_{Ref}$.
   * $\mathcal{L}_{\Psi}$ is not even defined
   * The KL term in Eq (3) is not defined (KL between what?! presumably, between the encoders' output distribution and the prior).
   * $\omega$ is not defined in Eq (3)
   *  No regularization coefficient for the losses?! (What if the shape was not smooth, and you are enforcing Laplacian loss for smoothness!)
   * Eq (4) is wrong! $x'\in \mathbb{R}^3$ is reserved for the reconstructed points after domain transfer in the latent space (i.e., after mapping to the target). I assume a $d$-dimensional latent space, where node embeddings are represented with $z$. Then the generator $G$ is defined on the latent space, i.e., $\mathbb{R}^d$, and must receive $z$ and not $x$ or $x'$.  Did the authors mean $G(E(x'))$?



**Paper Type:**

methodological development

**Questions To Address In The Rebuttal:**


* The paper could significantly benefit from an architecture figure to describe the approach pictorially

* Please revise your manuscript for a better flow of ideas and a better explanation of the proposed method.

* Please refer to the detailed comments to correct the mistakes in Section 2.3.

* Additional experiments could help the paper, and give the reader more confidence about the proposed approach.

* Balancing loss terms would be a challenge, and is completely ignored in the paper.

* The `prior-hole' problem in auto-encoder-based generative models has recently attracted much attention. A discussion on it and why it is or it is not a problem in the proposed approach is missing from the paper.

---

### Meta-Review · Area_Chair_vy8r · 2023-02-25

**Recommendation:** Accept (Poster)
**Confidence:** 4

**Metareview:**

The paper proposed a  generative model for generating virtual left ventricle shapes represented as graphs. The method utilizes a Beta Variational Auto-Encoder (VAE) with additional regularizations to match an input shape to a template shape in the latent space of the
beta-VAE, therefore implicitly solving the shape correspondence problem in the latent space and eliminating the need for initial shape correspondence.The authors numerically evaluate the proposed method on modeling left ventricle shapes and liver shapes. The reviewers initially reported several concerns, some of which were addressed by the authors during the rebuttal and therefore some reviewers improved their scores. In the light of these improvements, I suggest acceptance of the manuscript.